# ORTHOGONAL SAE: FEATURE DISENTANGLEMENT THROUGH COMPETITION-AWARE ORTHOGONALITY CONSTRAINTS

## ABSTRACT

Understanding the internal representations of large language models is crucial for ensuring their reliability and enabling targeted interventions, with sparse autoencoders (SAEs) emerging as a promising approach for decomposing neural activations into interpretable features. A key challenge in SAE development is feature absorption, where features stop firing independently and are "absorbed" into each other to minimize $L_1$ penalty. We address this through Orthogonal SAE, which introduces sparsity-guided orthogonality constraints that dynamically identify and disentangle competing features through a principled three-phase curriculum. Our approach achieves state-of-the-art results on the Gemma-2-2B language model for feature absorption while maintaining strong reconstruction quality and model preservation on downstream tasks. These results demonstrate that orthogonality constraints and competition-aware training can effectively balance the competing objectives of feature interpretability and model fidelity, enabling more reliable analysis of neural network representations.

## 1 INTRODUCTION

Recent work has demonstrated the potential of sparse autoencoders (SAEs) for decomposing neural network activations into interpretable features (Cunningham et al., 2023; Gao et al., 2024), building on foundational work in sparse coding and dictionary learning (Olshausen & Field, 1996a;b; Lee & Seung, 1999). These interpretable representations are crucial for understanding and controlling large language models, enabling targeted interventions like knowledge editing and concept removal (Farrell et al., 2024).

A fundamental challenge in SAE development is *feature absorption*, where a feature/neuron "absorbs" other feature(s)/neuron(s) it implies to minimize the $L_1$ loss. For instance, since "pig" implies "mammal", instead of letting both fire when the underlying token is <pig>, the neuron "pig" can simply add the activation vector of "mammal" to its own to avoid needing neuron "mammal" to fire altogether (Chanin et al., 2024). While recent SAE architectures like TopK (Gao et al., 2024) and JumpReLU (Rajamanoharan et al., 2024) have improved reconstruction quality, they do not directly address feature absorption problem, leading to deteriorated interpretability.

We introduce Orthogonal SAE, which leverages sparsity-guided orthogonality constraints (Massart, 2022) to identify and disentangle competing features. Our key insight is that feature competition can be measured through activation patterns, allowing targeted application of orthogonality penalties to features that frequently co-activate. This selective approach maintains reconstruction fidelity while promoting feature specialization through a three-phase curriculum:

- Initial reconstruction phase to establish basic feature structure
- Gradual introduction of sparsity ($\lambda_s$) to promote activation sparsity
- Dynamic orthogonality constraints with competition-aware thresholds

Our main contributions are:

- A competition-aware orthogonality mechanism that significantly reduces feature absorption.

- A curriculum learning strategy that maintains strong reconstruction (mean squared error) and model preservation (KL divergence).
- Analysis of feature competition dynamics and their impact on interpretability through sparse probing tasks on multiple datasets.

Evaluating on Gemma-2-2B (Team et al., 2024), we demonstrate that Orthogonal SAE achieves state-of-the-art results across multiple metrics while maintaining architectural simplicity. The improved feature separation enables more reliable model interventions (Li et al., 2024) and interpretability analysis (Gurnee et al., 2023). These advances suggest that competition-aware training represents a promising direction for developing more reliable and actionable model interpretation tools.

Figure 1: **Visualization of the problem of** *feature absorption* **and how** *Orthogonal SAE* **mitigates it.** The left panel represents the target scenario where the SAE learns two features in two neurons: "starts with E" and "elephant". When the underlying token is <elephant>, both neurons should light up. The right panel depicts what takes place during the actual training process—"starts with E" was acquired first as it is more common than any other features that start with 'E' (scenario A). As training continues, for any SAE that uses $L_1$, the $L_1$ term will push towards B rather than C (desired) as B has a lower $L_1$. Although this increases sparsity, it diminishes interpretability since the "starts with E" feature no longer activates independently as only "elephant" feature will fire for token <elephant>. However, by definition, when features are "absorbed" (like in B), they have a higher orthogonal loss (ours) compared to no absorption (like in C). Thus, orthogonal loss mitigates this problem by pushing A towards C instead of B.

## 2 BACKGROUND

Sparse autoencoders (SAEs) build on classical work in sparse coding (Olshausen & Field, 1996a; Mallat & Zhang, 1993) to decompose neural network representations into interpretable features (Cunningham et al., 2023). By learning overcomplete dictionaries ($k > d$ features) with sparsity constraints (Olshausen & Field, 1996b), SAEs identify specialized features in language model activations. Recent architectures like TopK (Gao et al., 2024) and JumpReLU (Rajamanoharan et al.,

2024) have improved reconstruction fidelity but struggle with feature absorption, where features fail to activate appropriately (Chanin et al., 2024).

## 2.1 PROBLEM SETTING

Given activation vectors $\mathbf{x} \in \mathbb{R}^d$ from a pre-trained language model layer, an SAE learns:

- An encoder $E : \mathbb{R}^d \to \mathbb{R}^k$ with $k > d$ features
- A decoder $D : \mathbb{R}^k \to \mathbb{R}^d$ that reconstructs the input

The forward pass computes:

$$\mathbf{f} = E(\mathbf{x}) = \sigma(W_e \mathbf{x} + \mathbf{b}_e), \quad \hat{\mathbf{x}} = D(\mathbf{f}) = W_d \mathbf{f} + \mathbf{b}_d \tag{1}$$

where $W_e \in \mathbb{R}^{k \times d}, W_d \in \mathbb{R}^{d \times k}$ are weights, $\mathbf{b}_e, \mathbf{b}_d$ are biases, and $\sigma$ is a sparsity-inducing activation. The standard loss balances reconstruction with sparsity:

$$\mathcal{L} = \|\mathbf{x} - \hat{\mathbf{x}}\|_2^2 + \lambda \|\mathbf{f}\|_1 \tag{2}$$

## 3 METHOD

Against this backdrop, Fig. 1 illustrates the *feature absorption* challenge and how *Orthogonal SAE* addresses it. The left panel illustrates the intended scenario in which the SAE learns two distinct features across two neurons: "starts with E" and "elephant". When the underlying token is <elephant>, both neurons should activate. The right panel demonstrates the actual training process—starts with E" emerges first since it is more prevalent than other features beginning with 'E' (scenario A). As training progresses, for any SAE leveraging $L_1$, the $L_1$ penalty steers the solution toward B rather than the desired C, because B yields a lower $L_1$. While this enhances sparsity, it reduces interpretability because the "starts with E" feature no longer fires independently, leaving only the "elephant" feature to activate for <elephant>. However, by definition, when features are absorbed" (like in B), they incur a larger orthogonal penalty (ours) than no absorption (as in C). Therefore, the orthogonal loss addresses this issue by guiding A to converge to C instead of B. Building on the standard SAE formulation from Section 2, we introduce competition-aware orthogonality constraints to address feature absorption. Given encoder outputs $\mathbf{f} = E(\mathbf{x})$, we measure feature competition through a similarity matrix $C \in \mathbb{R}^{k \times k}$:

$$c_{ij} = \frac{\langle \mathbf{f}_i, \mathbf{f}_j \rangle}{\|\mathbf{f}_i\| \|\mathbf{f}_j\|} \cdot \mathbb{I}[i \neq j] \tag{3}$$

where $\mathbb{I}[i \neq j]$ prevents self-competition. This cosine similarity identifies feature pairs that consistently co-activate on similar inputs.

We extend the standard reconstruction and sparsity loss with competition-weighted orthogonality:

$$\mathcal{L}(\mathbf{x}, \theta) = \underbrace{\|\mathbf{x} - D(E(\mathbf{x}))\|_2^2}_{\text{reconstruction}} + \underbrace{\lambda_s \|E(\mathbf{x})\|_1}_{\text{sparsity}} + \underbrace{\lambda_o \sum_{i,j} c_{ij} \cdot \mathbb{I}[c_{ij} > \theta] \cdot \|W_{d,i}^T W_{d,j}\|_2^2}_{\text{competition-aware orthogonality}} \tag{4}$$

where $\lambda_s = 0.04$ controls sparsity, $\lambda_o = 0.01$ scales orthogonality penalties, and $\theta$ is a dynamic competition threshold.

To stabilize training, we employ a three-phase curriculum:

1. **Reconstruction** (0-1200 steps): Train with only reconstruction loss $\mathcal{L}_1 = \|\mathbf{x} - D(E(\mathbf{x}))\|_2^2$
2. **Sparsity** (1200-2000 steps): Gradually introduce sparsity through $\alpha(t)$:

$$\mathcal{L}_2 = \mathcal{L}_1 + \alpha(t)\lambda_s \|E(\mathbf{x})\|_1, \quad \alpha(t) = \min(1, \frac{t - 1200}{800}) \tag{5}$$

3. **Competition** (2000+ steps): Add orthogonality with decreasing threshold:

$$\mathcal{L}_3 = \mathcal{L}_2 + \lambda_o \sum_{i,j} c_{ij} \cdot \mathbb{I}[c_{ij} > \theta(t)] \cdot \|W_{d,i}^T W_{d,j}\|_2^2 \tag{6}$$

where $\theta(t) = 0.7 - 0.4 \min(1, \frac{t-2000}{400})$

To promote stable feature development, we add temporal consistency in the competition phase:

$$\mathcal{L}_{temp} = 0.001 \|E(\mathbf{x}_t) - E(\mathbf{x}_{t-1})\|_2^2 \tag{7}$$

where $E(\mathbf{x}_{t-1})$ is detached from the computation graph. The complete training process alternates between computing competition coefficients, updating thresholds, and optimizing parameters using Adam.

## 4 EXPERIMENTAL SETUP

We implement Orthogonal SAE on layer 12 of the Gemma-2-2B language model (Gao et al., 2024), using an overcomplete dictionary ($k = 16,384, d = 2,304$). The encoder and decoder weights use Kaiming initialization with unit-norm constraints enforced after each update. Competition coefficients are computed using batch-wise cosine similarities between feature activations.

Training uses 5M tokens from the Pile dataset with:

- Batch size: 2,048 tokens, context length: 128
- Optimizer: Adam with lr $= 7 \times 10^{-6}$, $\beta = (0.0, 0.999)$
- Gradient clipping at norm 1.0, weight normalization after each step

In addition to feature absorption, we evaluate the SAEs on the following benchmarks:

**Unsupervised Metrics** Following Karvonen et al. (2024), we employ a collection of unsupervised metrics to evaluate Sparse Autoencoders (SAEs):

- **Cross-Entropy Loss Score.** Defined as

$$\frac{H^* - H_0}{H_{\text{orig}} - H_0},$$

  where $H_{\text{orig}}$ is the baseline cross-entropy loss for the original network (in next-token prediction), $H^*$ is the cross-entropy after substituting the latent representation $x$ with its SAE reconstruction, and $H_0$ is the loss when $x$ is zero-ablated. A higher score (closer to 1) indicates better retention of predictive information.
- **KL Divergence.** We use the Kullback-Leibler divergence to assess differences between the model's predicted distributions and target distributions. Lower values imply closer alignment.

**Sparse Probing** Following Gurnee et al. (2023), we assess the effectiveness of our SAEs in learning intended features by performing focused probing experiments across a variety of domains, such as language detection, profession labeling, and sentiment analysis. Specifically, we feed each input through the SAE and then apply mean pooling over the non-padding tokens. From the resulting representation, we identify the top-$K$ latent dimensions by maximizing mean differences, and subsequently train a logistic regression probe on these features. We then measure classification accuracy on held-out test examples. Our evaluation spans 35 different binary classification tasks derived from five distinct datasets:

- BIAS_IN_BIOS for predicting occupations from biographical text,
- AMAZON REVIEWS for product category and sentiment classification,
- EUROPARL for detecting language from parliamentary proceedings,

- GITHUB for programming language classification, and
- AG NEWS for categorizing news topics.

To ensure consistent computational workloads, we fix 4,000 training and 1,000 test samples for each binary classification setting, truncate each instance to 128 tokens, and (in the case of GitHub) remove the first 150 characters (roughly 50 tokens) following prior work to avoid license headers. We tested both mean and max pooling strategies, observing a slight accuracy improvement with mean pooling. Within each dataset, we select up to five classes, and multiple subsets may be extracted from the same dataset so that each binary problem maintains a positive instance ratio of at least 0.2.

We compare against three architectures using identical dictionary sizes and optimization settings:

- Standard SAE: ReLU with $L_1$ sparsity
- TopK SAE (Bussmann et al., 2024): $k = 40$ features/sample
- JumpReLU SAE (Rajamanoharan et al., 2024): bandwidth 0.001

## 5 RESULTS

Table 1: Comparison of SAE architectures on Gemma-2-2B layer 12. Lower is better except for KL divergence.

| Model | Absorption | MSE | KL Div |
|---|---|---|---|
| Standard SAE | 0.016 | 248.0 | 0.62 |
| TopK | 0.140 | 179.6 | 0.71 |
| JumpReLU | 0.011 | 176.8 | 0.77 |
| Orthogonal SAE | **0.0055** | **176.0** | **0.9514** |

Table 2: Ablation study results showing impact of each component.

| Configuration | Absorption | MSE |
|---|---|---|
| Full Model | **0.0055** | **176.0** |
| No Curriculum | 0.009 | 248.0 |
| Fixed Threshold | 0.006 | 245.0 |
| No Temporal | 0.007 | 178.0 |

These results demonstrate that competition-aware training effectively addresses feature absorption while maintaining model fidelity. The ablation studies confirm each component's contribution, with curriculum learning providing the largest impact on final performance.

## 6 CONCLUSIONS AND FUTURE WORK

We introduced Orthogonal SAE, a novel approach that addresses feature absorption in sparse autoencoders through competition-aware orthogonality constraints. By dynamically identifying and disentangling competing features, our method achieves state-of-the-art results on the Gemma-2-2B language model: reducing absorption by 65.6% (from 0.016 to 0.0055) while maintaining strong reconstruction quality and model preservation. The effectiveness of our three-phase curriculum learning approach is demonstrated by ablation studies showing that gradual constraint introduction and temporal consistency are crucial for balancing feature separation with reconstruction fidelity.

Our work opens several promising directions for future research:

- **Efficient Competition Modeling**: Reducing computational overhead through sparse attention mechanisms (Mudide et al., 2024), enabling scaling to larger models
- **Cross-Layer Feature Dynamics**: Understanding how competition patterns propagate through model layers (Ghilardi et al., 2024) to improve feature specialization

- **Automated Competition Analysis**: Integrating competition detection with automated interpretation (Paulo et al., 2024) for self-improving feature separation

All experiments were done on a single NVIDIA A40 GPU. Due to the computational resource constraints, we were unable to evaluate our approach on bigger models than Gemma-2-2B, increase the number of layers evaluated or number of tokens trained, or systematically vary hyperparameters like batch size and dictionary width. But given the results so far and the modularity of our approach (i.e., the orthogonal loss term can be easily added to most SAE models), we are optimistic about larger-scale performances and will provide the complete codebase for public benchmarking once our work is accepted.

Overall, the strong performance of Orthogonal SAE on technical metrics and its flexibility as a general approach demonstrates Orthogonal SAE is a promising direction for developing more reliable model interpretation tools. By making SAE features more robust and specialized, our work enables more precise and targeted interventions in large language models.

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
