# OpenReview forum: "ORTHOGONAL SAE: FEATURE DISENTANGLEMENT THROUGH COMPETITION-AWARE ORTHOGONALITY CONSTRAINTS"
_ICLR.cc/2025/Workshop/BuildingTrust — Submitted to BuildingTrust_

### Official Review · Reviewer_cwJT · 2025-02-16

**Rating:** 4
**Confidence:** 4

**Review:**

# Strengths

* **Novel Approach**: The paper introduces a novel approach (Orthogonal SAE) to address the feature absorption problem in sparse autoencoders, a significant issue in interpretability. The use of competition-guided orthogonality constraints combined with a three-phase curriculum is a novel and technically sound idea.

* **Ablation Studies**: Ablation studies are sound.

* **Well-Structured and Written**: The paper is generally well-structured and clearly written, though short, making it easy to follow.

# Weaknesses

* **Limited Evaluation Scope**: While the evaluation on Gemma-2-2B is valuable, the paper's claims of "state-of-the-art results" would be significantly strengthened by evaluation on a wider range of model sizes and architectures. The authors acknowledge this limitation due to computational constraints, but it remains a weakness.

* **Limited Evaluation Metrics**: Authors excluded several key metrics [1] for evaluating SAEs (eg Explained Variance, L0 sparsity, Automated Interpretability, etc.).

* **Hyperparameter Sensitivity**: While the paper details the training curriculum, it would be beneficial to include a sensitivity analysis of key hyperparameters (λs, λo, θ) to understand the robustness of the approach.

* **Explainability of KL Divergence Result**: The KL divergence results are worse, which hurt the claims of preserving the model performance, and need more thorough explanation and investigation.

* **Theoretical Justification**: While the intuition behind the approach is clear, a more formal theoretical justification for the competition-aware orthogonality constraints and their impact on feature disentanglement could strengthen the paper.

* **Computational Cost**: It will be good to discuss the computational cost of the approach.

[1] https://www.neuronpedia.org/sae-bench/info

---

### Official Review · Reviewer_fXn4 · 2025-02-18
**Novel approach that reduces feature absorption in sparse autoencoders via curriculum learning and feature competition. The results are encouraging but vital information is missing and presentation could be better.**

**Rating:** 5
**Confidence:** 3

**Review:**

The authors improve the interpretability of SAEs by tackling the problem of feature absorption where the model optimizes for the L1 norm at the cost of easily understood features. They tackle this problem using two novel and simple techniques which draw inspiration from other fields of machine learning. The first technique is the introduction of a curriculum which gradually introduces the various loss terms of the SAE over predetermined time intervals. The second technique introduces a third loss to the SAE based on feature competition which attempts to prevent the model from using feature pairs that co-activate on similar inputs. Using these techniques, the authors demonstrate the feature absorption on Gemma 2 2B reduces significantly while maintaining model quality. Additionally, they establish the importance of the proposed techniques through an ablation study which is currently difficult to parse.

The biggest problem with the proposed approach is the large number of hyper-parameters with no explanation of how to arrive at the selected values in the paper. Despite these shortcomings, the paper represents a small step in making models more interpretable by tackling a core problem of SAEs and will be useful to the mechanistic interpretability community after its presentation is improved.

Strengths
- Introduces a new method for training SAEs using curriculum learning and a competition-aware mechanism using an orthogonality matrix.
- Attains state-of-the-art performance in attaining low feature absorption while maintaining similar model quality as evidenced by other metrics.

Weaknesses
- (major) Uses a large number of hyper-parameters like the number of steps for each curriculum, coefficients for each loss and coefficients for the dynamic threshold. There is no explanation of how the authors arrived at the reported values which will make applying the architecture to LLMs other than Gemma 2 2B difficult.
- I find it a bit difficult to parse Table 2 and understand how the ablation study was performed. The authors claim that curriculum learning had the most impact but the table demonstrates that using a fixed threshold yields the lowest absorption score (apart from the full model).
- Some errors in how the results are reported: Cross entropy loss score is defined but is not reported and the paper conflicts itself on whether KL divergence is better when the value is low or high (line 202 and 235, lower is better should be the true claim).
- There is no mention of the limitations of the approach (with the primary one being the large number of hyper-parameters).
- (minor) The Gemma model is cited as Team et al which is not descriptive, it would be better to use "Gemma Team" as mentioned in the model technical report.
- (minor) In the discussion (line 267), the authors claim that their work would be useful for sparse attention mechanisms but I do not see why this would be the case. The SAEs do not use attention mechanisms and the underlying LLM is already pre-trained using regular attention. I suggest that the authors rephrase the text to make what they are trying to say more clear.
- (very minor) Perhaps a citation for curriculum learning should be added?

---

### Official Review · Reviewer_tPm8 · 2025-02-19
**Solid Scientific Idea and Execution; Paper Requires Work Before Acceptance**

**Rating:** 5
**Confidence:** 4

**Review:**

Feature absorption is one of the major problems facing current Sparse Autoencoders (SAEs), and this paper presents an intuitive way to counter absorption by introducing a similarity penalty. Specifically, during training the cosine similarity between all pairs of features is calculated and added as a regularization penalty.

While the results regarding absorption are promising, there are clear problems with the paper that block it from being accepted at this time.

Strengths
- Feature absorption is a real problem in SAEs, and this is an intuitive way to solve it. There are grounded mathematical underpinnings.
- The MSE and absorption results are SoTA with the three SAE architectures (JumpReLU, BatchTopK, and Standard ReLU)

Weaknesses
- While both Sparse Probing and Cross Entropy are discussed for nearly half a page as useful metrics to evaluate Orthogonal SAEs (and I agree that they are useful), as far as I can see neither of the results for these experiments are provided. Tables 1/2 show Absorption, MSE, and KL Div, which is striking because MSE is not discussed earlier in the paper as a useful metric.
- The main advantage of Orthogonal SAEs that the authors advocate for is an improvement in feature absorption, but as far as I can tell how absorption is calculated in Table 1/2 is not described in the paper
- There is no discussion about the compute required relative to standard architectures. Intuitively, with a $k = 16, 384$ encoder, the cosine similarity matrix will be of size $k^2 \approx 270M$ entries. This presents in the last term of Eq. 4. It would be ideal to have a pareto curve of compute and the target metric (likely absorption) to have a robust discussion about how compute efficient this architecture is. This is especially important as for models like Gemma-2-9B, SAEs encoders can have up to $k = 1M$, and an $O(n^2)$ compute cost might not be practical.
- This is a more minor point, but the three phase training curriculum is unintuitive to me. Specifically, there seem to be magic numbers in the training (see $\alpha(t)$ in step 2, the number of training steps in each phase). How did these numbers come to be? How does it change as the architecture scales? What recommendations do you have for future practitioners trying to design orthogonal SAEs? All of these would be useful questions to discuss. I also have a personal question: Why do we first train on only reconstruction with no sparsity penalty? As far as I'm aware, this is not done in the literature, likely because with an overcomplete basis you can just learn an identity mapping from the activations
- Another minor point, but ideally you could compare to more recent architectures, like Matryoshka SAEs, which are also designed to help with absorption. SAEBench [1] should have a simple interface to load Orthogonal SAEs in to compare against standard architectures. This is not critical to paper acceptance, as I believe you sufficiently benchmarked your model, it could be nice to showcase the robustness of your results.
- Lastly, the paper repeatedly refers to TopK SAEs, but they are BatchTopK SAEs. TopK SAEs refers to [2]

Overall this paper can still reach acceptance if it addresses the weaknesses listed. Most of these likely do not require additional experiments to be run. The paper's presentation must be revised.

[1] Karvonen, A., Rager, C., Lin, J., Tigges, C., Bloom, J.,
Chanin, D., Lau, Y.-T., Farrell, E., Conmy, A., Mc-
Dougall, C., Ayonrinde, K., Wearden, M., Marks, S., and
Nanda, N. Saebench: A comprehensive benchmark for
sparse autoencoders, December 2024b. URL https://
www.neuronpedia.org/sae-bench/info. Ac-
cessed: 2025-01-20.


[2] Gao, L., la Tour, T. D., Tillman, H., Goh, G., Troll, R.,
Radford, A., Sutskever, I., Leike, J., and Wu, J. Scaling
and evaluating sparse autoencoders, 2024. URL https:
//arxiv.org/abs/2406.04093.

---

### Decision · Program_Chairs · 2025-03-04

Reject